# Rule-based outlier detection of AI-generated anatomy segmentations

Deepa Krishnaswamy[1], Vamsi Krishna Thiriveedhi[1], Cosmin Ciausu[1], David Clunie[2], Steve Pieper[3], Ron Kikinis[1], and Andrey Fedorov[1]

[1]Brigham and Women's Hospital, Boston, MA, USA
[2]PixelMed Publishing, Bangor, PA, USA
[3]Isomics, Cambridge, MA, USA

## Abstract

There is a dire need for medical imaging datasets with accompanying annotations to perform downstream patient analysis. However, it is difficult to manually generate these annotations, due to the time-consuming nature, and the variability in clinical conventions. Artificial intelligence has been adopted in the field as a potential method to annotate these large datasets, however, a lack of expert annotations or ground truth can inhibit the adoption of these annotations. We recently made a dataset publicly available including annotations and extracted features of up to 104 organs for the National Lung Screening Trial using the TotalSegmentator method. However, the released dataset does not include expert-derived annotations or an assessment of the accuracy of the segmentations, limiting its usefulness. We propose the development of heuristics to assess the quality of the segmentations, providing methods to measure the consistency of the annotations and a comparison of results to the literature. We make our code and related materials publicly available at `https://github.com/ImagingDataCommons/CloudSegmentatorResults` and interactive tools at `https://huggingface.co/spaces/ImagingDataCommons/CloudSegmentatorResults`.

## 1 Introduction

Availability of annotations is critical for secondary use of public imaging datasets. In the case of large-scale datasets, annotating them manually in a timely manner is not feasible. In recent years, artificial intelligence (AI) tools have shown promise in automatic annotation of both anatomy and pathology, for a variety of use cases. For instance, the TotalSegmentator model can annotate up to 118 anatomical structures in computed tomography (CT) volumes [1] and has recently been extended to magnetic resonance imaging (MRI) [2]. Other popular models include the nnU-Net framework [3] and its set of pre-trained models covering segmentation of the prostate, kidneys, cardiac substructures, and other regions, for multiple modalities. More recently, the SegVol [4] method provides a universal, interactive method for annotating medical images.

Despite the availability of robust pre-trained models, there are still many large publicly available datasets without sufficient annotations. One of the largest collections in the NCI Imaging Data Commons (IDC) repository [5] is data from the National Lung Screening Trial (NLST) [6]. The CT arm of this collection holds over 26K patients scanned over a period of three years, yielding over 200K computed tomography (CT) volumes. Until recently, most of these were unlabeled, complicating their downstream use. Earlier, we used the TotalSegmentator pre-trained model to volumetrically segment the organs and anatomic structures in over 125K of those CT images, and extracted 28 radiomics

38th Conference on Neural Information Processing Systems (NeurIPS 2024).

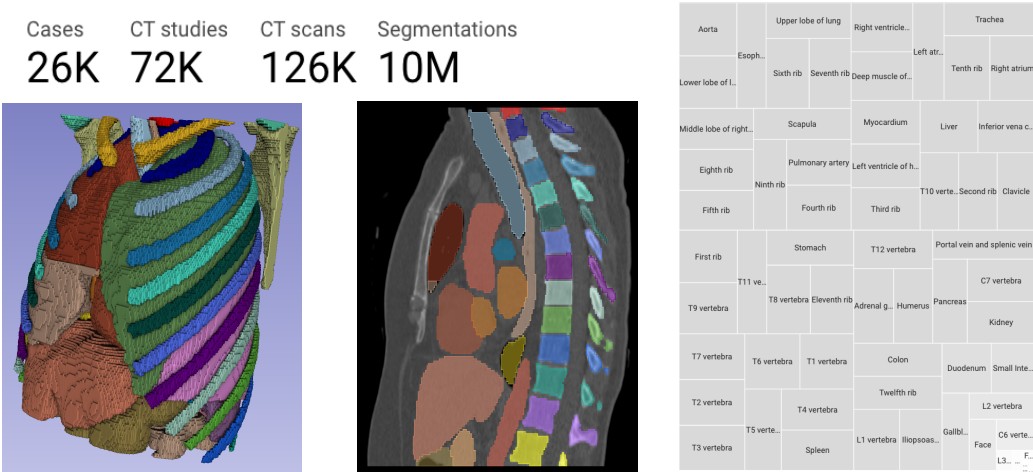

Figure 1: Example segmentations generated from the TotalSegmentator AI model on a patient from the National Lung Screening Trial cohort, displayed using 3DSlicer [15]. The treemap summarizes anatomic structures segmented across the entire cohort.

features for each of the 9.5 million generated segmentations [7, 8]. The annotations generated by this AI-based curation effort are publicly available from IDC.

A practical challenge in the downstream use of AI-generated segmentations is in the lack of certainty about their correctness. The NLST collection does not include ground truth segmentations. While selective visual checks can help build confidence about these annotations, with the total number of over 9.5 million of segmentations and a range of 23-96 segmented structures per individual scan, neither complete review by the experts nor the creation of ground truth for a sample of this dataset is practical. Additionally, it would be beneficial to inform the user of segmentations that may be problematic. As our dataset contains annotations for over 125K+ CT volumes, it is virtually impossible to perform a manual review without the aid of efficient methods for detecting failures.

Several methods have been developed for the analysis of segmentation results without associated expert-derived or ground truth data. These can be divided into two main approaches, ones that focus on predicting overlap metrics [9–11], and ones that predict the error-specific regions of a segmentation mask [12–14]. However, all of these machine learning and deep learning approaches suffer from limitations. For instance, they may exhibit low performance when evaluated on data with a different domain from the training dataset. Additionally, the fact that all methods involve training one or more models can be time-consuming.

We therefore propose simple heuristics that aim to help with detecting failures and assessing the performance of the volumetric segmentations. This approach does not require the use of machine learning and may potentially be beneficial beyond the dataset evaluated therein. Given the AI-generated segmentations of over 9.5M structures across 125K+ CT volumes of the NLST collection, we evaluate the developed heuristics and provide interactive tools for visualizing and exploring analysis results, and assessing the presented heuristics. As a surrogate of the effectiveness of those heuristics, we investigated if removal of the results deemed to be failures by the heuristics leads to improved consistency of 1) volumes of left vs right structures such as the ribs 2) within-patient volumes of structures, and 3) vertebral volumes compared to a population in the literature. Additionally, to enable user interaction and exploration of the AI-generated segmentations, we propose multiple tools the user can utilize for benchmarking purposes. Finally, we make all of our code and interactive tools publicly available on `https://github.com/ImagingDataCommons/CloudSegmentatorResults` and `https://huggingface.co/spaces/ImagingDataCommons/CloudSegmentatorResults`.

## 2   Methodology

We developed four heuristics for analyzing AI-generated results and identifying problematic segmentations in the absence of expert annotations. Figure 1 displays an overview of the segmentations that we generate using TotalSegmentator, displayed using 3DSlicer [15]. These heuristics are in part based upon the DICOM metadata extracted from the segmentations presented in previous work [7, 8]. Please see the Supplementary materials for details concerning the metadata. Using the set of heuristics, we investigated the effect of them on three studies: 1. left vs right volumes of the ribs, 2. within-patient volumes, and 3. comparison of AI-generated vertebral volumes to a population study.

### 2.1   Heuristics

The four heuristics are described below. Barring the connected components check, the remaining heuristics yields a true/false flag for each of the individual segments that need to be analyzed, with the "fail" value corresponding to a segment flagged as problematic by a given heuristic rule. We left the connected components in numerical form to let the user adjust the threshold if needed.

1. Segmentation completeness: Depending on the inferior-to-superior extent of the axial CT scan, some anatomical structures may be included only partially. To remove these incomplete segmentations from further analysis, we evaluated the completeness of the segmentation by ensuring that there was at least one empty slice above and below the segment. Furthermore, we hypothesized that the segmentation might not be complete if it appears on the most inferior or superior transverse slices of the scan, indicating that it could extend beyond the scanned range, despite a theoretical possibility of a perfect alignment occurring on the terminal slices. We note that this heuristic can only indicate whether TotalSegmentator had the opportunity to segment a structure completely, but not the accuracy.

2. Connected component: Each anatomical region that is segmented volumetrically should be continuous and consist of a single connected component. However, using the pre-trained TotalSegmentator model with minimal post-processing can yield unconnected components for an anatomical region. Using the pyradiomics [16] package, the `VoxelNum` field proves the number of connected components, which in the ideal case should be one. We therefore used this field to identify segmentations with extraneous or noisy voxels.

3. Laterality: Segmentation algorithms may produce the incorrect laterality label (left vs right) of a region. To detect this, we evaluated the laterality by using metadata extracted from the segmentations using the pyradiomics [16] general feature `CenterOfMass` field. This attribute provides the center of mass of the region in the world-coordinate system. Using this system, the coordinates increase from right to left; therefore, we can easily determine if the laterality of a paired structure is correct.

4. Minimum volume from voxel summation: To our knowledge, the adrenal gland is the smallest organ segmented by TotalSegmentator. The average volume according to [17] is approximately 5 mL. We used the pyradiomics feature `Volume from Voxel Summation` to discern volume and chose 5 mL as the threshold for all segmentations to remove artifacts.

### 2.2   Interactive visualization

To facilitate analysis of the segmentation results, we developed an interactive dashboard based on the Streamlit framework (`https://github.com/streamlit/streamlit`) and hosted it on the free tier of Hugging Face spaces (`https://huggingface.co/spaces/ImagingDataCommons/CloudSegmentatorResults`). The dashboard consists of two pages. The 'Summary' page contains the results of applying each of the four heuristics to each of the segments. The table can be sorted to quickly gain insights into which of the segmentations were flagged as outliers. The "Plots" page features two types of plots and includes filtering options for radiomics feature, anatomical structure, laterality, and the four heuristics. We display upset plots, which show how many segments passed or failed the heuristics, and in what combinations. Additionally, we display violin plots, which demonstrate the distributions of the standard deviation of radiomics features before and after applying the heuristics within a patient. This helps in studying the effect of the heuristics on the consistency of a radiomics feature value distribution for a given anatomical structure within a patient. Both plots are updated dynamically depending on the choice of filters.

## 2.3 Left vs right volumes of the ribs

The ribs comprise a large portion of the regions segmented by TotalSegmentator, accounting for 24 out of the total 104 segments. However, the ribs may suffer from a number of problems which could result in inaccurate segmentations, for instance, they are relatively small structures. Secondly, training of the TotalSegmentator method utilized data from the RibFrac 2020 challenge [18] (`https://ribfrac.grand-challenge.org/dataset`) that provided a single segmentation of the ribs, which were then post-processed by the developers of TotalSegmentator into 24 individual segments. Lastly, the portion of the ribs close to the vertebrae was excluded in the original segmentation. Due to these potential issues and the symmetry of the ribs, we chose to focus on the consistency of the left vs right rib segmentation.

For each pair of left vs right ribs, we computed the normalized difference of the volume: (left-right)/(left+right). The differences were computed for four sets of volumes, the original before any filtering is applied, after the segmentation completeness is applied, after the single connected component is applied, and after the laterality heuristic is performed. Multiple linear effects models were used to assess the effect of the addition of each heuristic. In all cases, the patient ID was included as a random effect, to account for the fact that a patient is scanned multiple times.

## 2.4 Within-patient volumes

Each patient from the NLST study was scanned three times over three years. One or more scans were acquired within each time point (known as a study). Within each study, various convolution kernels were used for reconstruction of the CT scan, in order to enhance and visualize different parts of the chest. According to the NLST protocol [6], a single helical scan was obtained from a patient, and two or three axial reconstructions were performed. The patient was scanned until a satisfactory scan was obtained. Since each patient is scanned over three years, we expect some variability in the organs. Therefore, we chose to study the volumes of vertebrae before and after applying the heuristics, to see if there are significant differences between the distributions.

## 2.5 Comparison of AI-generated vertebral volumes to a population study

We chose to study the vertebra, as the vertebral segments account for approximately 23% of the possible structures segmented by TotalSegmentator. The volume of the vertebrae in particular is a useful measure for a number of different areas. For instance in osteoporosis, a disease that causes reduced mass of the bones and vertebral fractures, the volume of the vertebra may be used to monitor the progression of the disease [19]. We compared our observations with Limthongku et al. 2010 study [20] of the volume of the lumbar and thoracic vertebrae. In that study CT scans were performed on 40 patients (even distribution of men and women) and the BrainLab software (Munich, Germany) was used to calculate the vertebral volumes. Both interobserver and intraobserver studies were performed to confirm the reliability of the volume measurements.

# 3 Results

## 3.1 Heuristics and interactive visualization

The "Summary" page of the dashboard (`https://huggingface.co/spaces/ImagingDataCommons/CloudSegmentatorResults)`) reveals how many segments corresponding to the individual anatomic structures passed each of the heuristics when applied independently.

1. Segmentation Completeness: Given that the region of interest in the NLST scans is the chest area, most organs within the thoracic region were segmented completely. The segmentation performance was highest for organs located in the middle of the thoracic region and gradually decreased for organs located towards the outer regions. Only 32% (34/104) of the total organs segmented by TotalSegmentator passed the segmentation completeness check in over 90% of the cumulative total number of segmentations. These successfully segmented organs included 8 thoracic vertebrae (T3-T10), 14 ribs (two through eight), all 5 lung lobes, all 4 heart chambers (atria and ventricles), the pulmonary artery, and the myocardium.

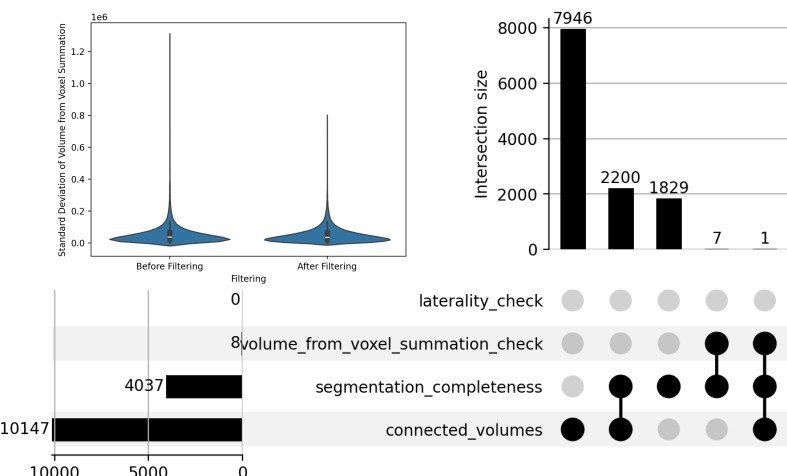

Figure 2: Picking volume from voxel summation as the radiomics feature, and selecting upper lobe of lung as the body part and left as the laterality, upset plots for failed checks indicate a total of 10,147 segmentations had more than one connected volumes, 4037 segmentations did not pass the segmentation completeness check, and 8 segmentations did not the minimum volume of 5 mL criteria. Intersection sizes on the top right reveal how these were distributed in each combination. Furthermore, selecting segmentation completeness to pass, the violin plot shows the change in distribution of standard deviation of volume from voxel summation feature before and after applying the connected volumes filter.

2. Laterality : Laterality test was performed for all paired organs, a total of 56 organs. TotalSegmentator assigned laterality exceptionally well, with nearly 100% accuracy in 75% (42/56) of the organs segmented, and over 95% accuracy in 98% (55/56) organs.

3. Connected components: TotalSegmentator uses nnU-Net as the base algorithm, performing segmentations volumetrically. Ideally, we expect a single connected volume per segmentation. All four heart chambers, the pulmonary artery, aorta, myocardium, and deep back muscles had more than 98% of segmentations with a single connected volume. The portal and splenic veins had the lowest number of single volumes, with only 13% (16385/124329) of segmentations being single volumes. All vertebrae except for T2, T3, and T7-12 had less than 80% of segmentations with a single volume. Specifically, C7, L3, and L4 had less than 50% of segmentations with a single volume.

4. Minimum volume from voxel summation: We found that 58% (60/104) of the organs had 90% or more segmentations with volume of at least 5 mL. As anticipated, the majority of these organs were located in the thoracic region. However, it is noteworthy that the eleventh and twelfth ribs, all lumbar vertebrae (with the exception of L1), and all cervical vertebrae had less than 90% of their segmentations meeting the minimum volume threshold. All organs below abdomen area failed the threshold as well.

In the following we perform a more detailed analysis of the results to evaluate whether they have effect on quantitative feature analysis. Figure 2 displays an example visualization from the dashboard.

## 3.2 Left vs right volumes of the ribs

We first investigated the effect of applying multiple heuristics to aid in the filtering of series of interest, as demonstrated in Figure 3. We assign the heuristics to the following: A = segmentation completeness check, B = number of connected components = 1, C = volume > 5 mL, D = laterality check, and apply them successively. For each rib, we calculate the normalized difference between the left and the right rib by computing (left-right)/(left+right). In the figure, we plot the mean normalized difference for each rib pair, and a line connecting the mean +/- one standard deviation. We observe that the application of each successive heuristic usually decreases the mean normalized difference,

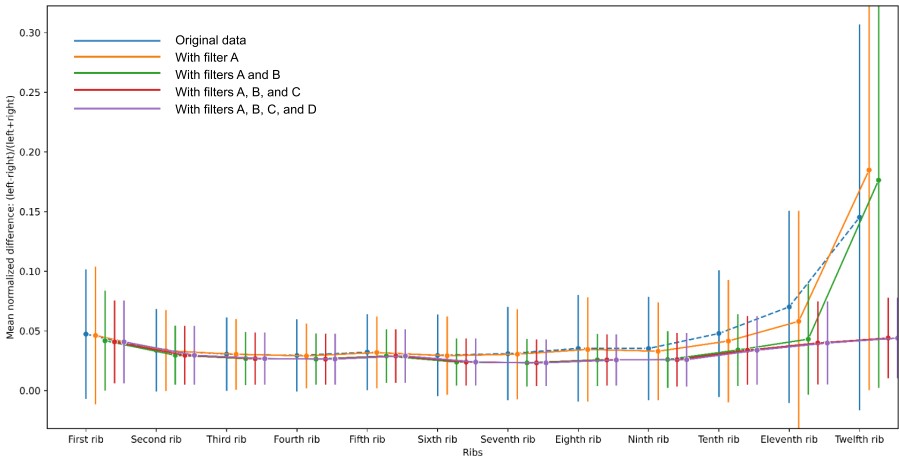

Figure 3: We plot for each pair of ribs the mean normalized volume difference (left-right)/(left+right). We also include the mean +/- the standard deviation. Each line for each rib pair corresponds to the inclusion of successive heuristics, starting from the original data and applying filters. Here, A = segmentation completeness check, B = number of connected components = 1, C = volume > 5 mL, D = laterality check. We observe a decrease in standard deviation as heuristics are applied for most ribs, indicating the removal of outliers.

which is satisfactory as we only want to include rib pairs that are symmetric and have a similar volume between the left and the right. We also observe that for most ribs, the application of successive heuristics lowers the standard deviation of the normalized difference, also indicating that outliers are removed from this process. We performed statistical testing to evaluate the effect of applying each of the successive heuristics, and to see if there was a significant difference between the original data and after applying all of the heuristics. Linear mixed-effects modeling was performed between each set of data and a sucessive heuristic applied, taking into account that each patient had repeated measures by considering it to be a random effect. Detailed results can be seen in the supplementary materials, where we observed the significant effect of the segmentation completeness heuristic.

There are cases where the heuristics worked as expected, and removed problematic series. For instance, in Figure 4A, the heuristic correctly identified segmentations that were incomplete, as can be seen for the 12th rib. As the scan does not fully cover the 12th rib, this series was removed from further analysis. However, there are cases where the heuristics did not work as expected. In Figure 4B, we can see an example of one of the series not filtered. The 12th rib is incorrectly labeled, as part of the vertebrae is likely segmented instead. Additionally, it can be seen that the 11th rib is oversegmented and incorrectly labeled, as part of it should be the 12th rib.

### 3.3 Within-patient volumes

Figure 5 demonstrates the within-patient consistency of the right kidney. For each patient, we compute the standard deviation of the volumes, and plot the distribution of these standard deviations. We compare the original volumes (no heuristics applied) to the distribution of volumes after all heuristics have been applied. We observe a lower standard deviation of volumes after filtering, as well as a larger concentration around the lower median, indicating that filtering likely removed some problematic series. However, despite appying the heuristics, they do not remove all outliers as observed in Figure 5.

### 3.4 Comparison of AI-generated vertebral volumes to a population study

In Figure 6 we plot on the left the distribution of the number of series per vertebra. We observe that there is a significant drop off in the number of series for the cervical and lumbar vertebrae, which is appropriate as the NLST cohort is for imaging the chest where the thoracic vertebrae are located. Therefore, we focus on extracting the volume features of only the thoracic vertebrae, as seen on the

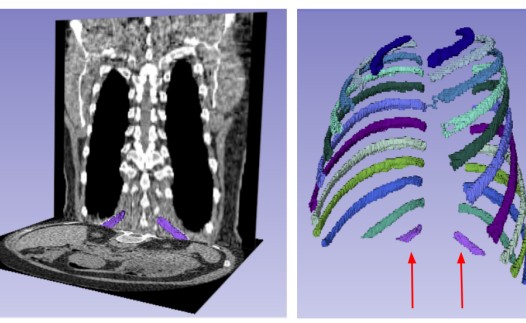
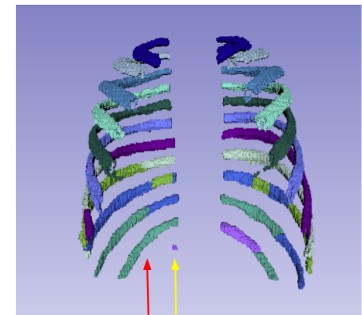

Figure 4: (Left) Example of an outlier that was correctly removed with our heuristics, created using 3DSlicer [15]. The ribs are not completely segmented (indicated by the red arrows), as the scan is cut off. (Right) Example of an outlier that was not removed with the heuristics, created using 3DSlicer [15]. Part of the vertebra is mislabeled as the 12th rib (yellow arrow), and part of the 11th rib should be the 12th rib (red arrow).

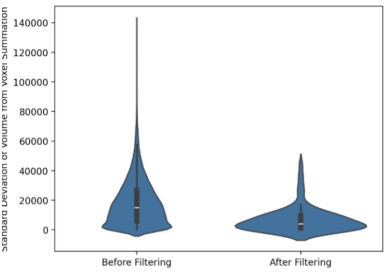
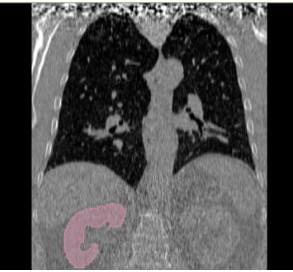
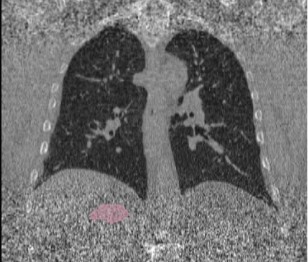

Figure 5: Within-patient consistency of the right kidney before and after applying heuristics. The figure in the middle shows an example of a kidney that passed all heuristics, while the figure on the right also passed all heuristics but should not have been included in the analysis.

right. Here we can observe the distribution of volumes after applying the heuristics, and an increase in the median volume as one goes from superior to inferior, which agrees with the literature [20].

We also compared the volume values after the four heuristics to the literature, as seen in Figure 7. We observe a similar trend in the thoracic vertebrae for our population vs the paper for both males and females. However, we do observe a large shift in the volumes from our approach vs that of the paper. This is due to the fact that the study from the paper measured the volume of the vertebral body, while our method measured the volume of the entire vertebra, which consists of both the vertebral body (anterior arch) and the posterior arch.

## 4   Discussion

We have provided a set of heuristics that can be applied to help identify problematic segmentations in large datasets. With our dataset consisting of over 125K CT volumes and up to 104 possible regions segmented in each of those CT volumes, the task of manual evaluation, and identifying failures is next to impossible. In this manuscript we have demonstrated several possibilities of not only how one can identify failures in the segmentation, but how one can quickly summarize datasets and annotations, and perform a comparison to literature. Overall, our provided dataset and annotations enhance further exploration beyond the use cases demonstrated in this paper. Additionally, we make our code and

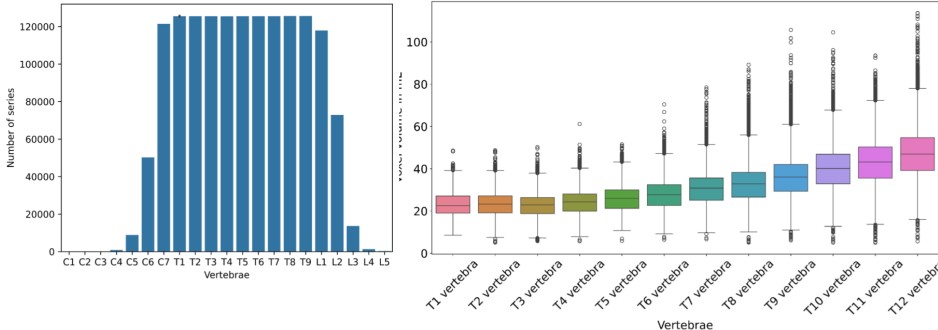

Figure 6: On the left we show the distribution of the original number of series per vertebra, and observe the decrease in series in the cervical and lumbar regions. On the right we plot the distributions of the volumes (mL) of the thoracic vertebrae after applying the four heuristics, and observe an increase in the mean volume which follows literature [20].

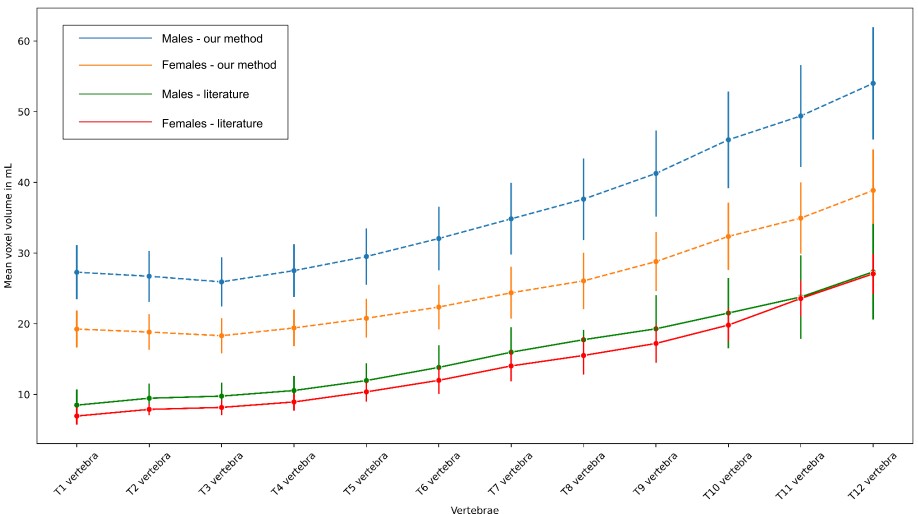

Figure 7: Here we plot a comparison between the literature and our vertebral volumes (after applying heuristics) for males vs females. The males from the literature are in green, with ours in blue, and the females from literature are in red, with ours in orange. We also plot the mean and the line to +/- 1 standard deviation. We can observe the similar trend in both the males in females in comparison to the literature, with the shift in volume occurring because different parts of the vertebra were measured.

data available in order to encourage reproducibility, and demonstrate how the same qualitative and quantitative heuristics can be used for analysis of any similar data.

There are several applications where the developed heuristics can have immediate impact. First, by detecting the problematic cases we can reduce the noise in the data for studies that are relying on segmentation-derived measurements. Second, by eliminating failed segmentations, the burden of manual review of the AI-generated segmentations can be reduced. This preliminary evaluation ensures that only the most accurate and reliable segmentations are forwarded for manual review.

The heuristics that we developed do suffer from a few limitations. The segmentation completeness heuristic requires an unsegmented slice above and below the segmented region. Segmentation of a single voxel with the empty slices above and below would be qualified as complete. Additionally, the segmentation completeness check may only be useful for certain organs, for instance ones that are more localized in the inferior-superior direction. For organs such as the vessels or veins, or deep muscles of the back, that have a much larger extent in the inferior-superior direction, this type of

heuristic may not be beneficial. For the laterality check, we observed that this heuristic did not make a significant difference in assessing failures of the segmentations. However, this is likely due to the robustness and high performance of the TotalSegmentator algorithm. For other algorithms that are either not as robust or are in development, this metric could be potentially useful.

For the heuristic that treats having a single connected component as ideal, there are a few limitations that exist. TotalSegmentator sometimes produces artifacts in many segments. For example, a single connected component could consist of just a single voxel and would not be flagged by the check. On the other hand, a segmentation with two components may be flagged as problematic even if the first component corresponds to a precise segmentation and the second component containing a single voxel. Other limitations include inability to identify cases where one segment is mislabeled as another, or where a segmentation is technically correct, but the entirety of the structure is not segmented. Finally, none of the heuristics we introduced are able to assess the alignment of the boundary of the segmentation with the boundary of the organ/structure in the image. These heuristics cannot replace quantitative assessment of the overlap with the expert segmentations.

There are also some limitations in the analysis that we performed. In the evaluation of the vertebral volume distributions we compared our results with those by Limthongkul et al.[20], which reported the distribution of the volumes of the vertebral body (anterior arch). Our analysis is based on the entire vertebra segmentation, which consists of both the vertebral body (anterior arch) and the posterior arch. Therefore, we could not perform direct comparison with the results of that study. Another limitation of our analyses was the issue of repeated measures. Each patient was scanned multiple times (multiple studies). Within each study, each patient consisted of both series reconstructed from the same scan, and new scans. Our statistical analysis did not account for all the hierarchical degrees of repeated measures, and instead only considered the top level patient-wise repeated measures. Additionally, we did not perform an analysis of how our heuristics perform with regards to gender and race, and if these heuristics unfairly impact specific groups.

The dashboard we developed allows users to quickly filter for regions of interest and visualize the effect of applying different heuristics. However, with more time, there are additional features that would be useful. For instance, we currently can only select a single region at a time, along with a single feature. This limits the amount of comparison that the user can currently perform. Additionally, one may want to compare features for the left vs right structures, which is currently not supported. Improvements for filtering in the table by body part, for instance picking all rib regions, would be helpful.

There are numerous improvements that can be implemented. A portion of our heuristics are dependent on thresholds, such as the volume of the segmented area. Those could be replaced by structure specific values from the literature. Additionally, the heuristics could include additional radiomics features among those available. We would also like to perform more population studies for regions that are associated with the lungs and lung cancer, as those were not specifically a focus in the manuscript. We could also consider the development of AI or machine learning models to capture the uncertainty of the model and perform quality control, by leveraging the TotalSegmentator ground truth data for training. Additionally, the interactive Hugging Face dashboard could be improved in terms of comparing multiple regions and features, and including additional plots and more user-friendly filtering.

## 5  Conclusion

We have demonstrated the potential of simple heuristics to aid in the quality control and detection of failures in large, annotated datasets without access to ground truth. We have proposed the use of four heuristics to capture and detect issues with if a region is fully segmented, the presence of multiple components, the laterality of the region and the minimum volume from voxel summation. With these measures we have provided methods to easily interact with, summarize, and understand the data. Additionally, we have demonstrated three studies that can be performed with the use of these heuristics, including the studying of the consistency of left vs right rib segmentations, the variability of a region for a patient, and a comparison to literature. While the proposed approach cannot flag all of the problematic segmentation results and has limitations, it is relatively easy to apply and is effective in identifying some of the outliers. We hope our work presented here can stimulate future research into the challenging task of automating quality control for the AI-generated analysis results.

## Acknowledgments and Disclosure of Funding

This work was supported in part by NIH NCI under Task Order No. HHSN2611 0071 under Contract No. HHSN261201500003l. Our use of JetStream2 resources was supported by the ACCESS project 230025 [21, 22].

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
