# OpenReview forum: "Rule-based outlier detection of AI-generated anatomy segmentations"
_NeurIPS.cc/2024/Datasets_and_Benchmarks_Track — Submitted to NeurIPS 2024 Track Datasets and Benchmarks_

### Official Review · Reviewer_oSdE · 2024-07-18
**Out-lier detection of anatomy segmentation using human-defined rules**

**Rating:** 4
**Confidence:** 4
**Correctness:** Yes.
**Clarity:** Yes.

**Review:**

This paper is easy to understand. The authors propose several ways to detect outliers of AI-generated anatomy segmentation. Even though their task is interesting, I’m not sure this paper really fits on NeurIPS dataset and benchmark track because it does not propose any benchmarks or datasets. There are no comparison methods on this task. Rather, it seems like an analysis of the Total Segmentor dataset. If this paper includes some outlier detection benchmarks on AI-generated segmentation map and compare performance of existing approaches, its impact could be larger.
Another concern is that their proposed methods are not applicable to various types of outliers, though they could be useful. Each heuristic is expected to work properly for only a subset of organs. For instance, small organs or those with complex structures (like blood vessels) might not be effectively analyzed using the same heuristics as larger, simpler organs. To demonstrate the effectiveness of these heuristics across different types of organs, it is necessary to include results for small organs or distributed organs (e.g., hepatic vessel tumors). The authors also acknowledges that these thresholds may not capture all nuances of segmentation errors, especially in complex anatomical structures (Section 4, Discussion).

**Strengths:**

The paper is well written and clearly structured.
The study excels in devising strong heuristics for evaluating AI-generated segmentations without relying on ground truth annotations.
The study introduces four key heuristics—segmentation completeness, connected component analysis, laterality check, and minimum volume threshold—providing practical tools for initial quality control across extensive datasets.

**Additional Feedback:**

Please check my review.

**Documentation:**

Yes.

**Ethics:**

No.

**Limitations:**

Yes
The paper extensively discusses the limitations of the authors' work. It identifies challenges with specific heuristics like segmentation completeness, laterality checks, and single connected component detection. Methodological constraints, such as the inability to handle detailed hierarchical repeated measures and the absence of gender and race impact assessments, are also acknowledged. Furthermore, the paper notes user interface limitations in the interactive dashboard. Despite these shortcomings, it emphasizes the effectiveness of the heuristics in dataset enhancement and comparative analyses, while suggesting opportunities for future refinements and broader applications.

**Opportunities For Improvement:**

Please check my review

**Relation To Prior Work:**

Partially yes. Literatures on out-lier detection based on segmentation map should discussed.

**Summary And Contributions:**

The paper presents a set of heuristics designed to evaluate the quality of AI-generated medical imaging segmentations without requiring expert annotations. Generating manual annotations for medical imaging datasets is hindered by time-consuming nature and variations in clinical conventions. Artificial intelligence (AI) offers a potential solution but faces obstacles like the absence of expert annotations for validation. In this paper, they developed four specific heuristics to check segmentation completeness, connected components, laterality, and minimum volume. The study demonstrated that these heuristics improve data quality and reduce the need for manual review when applied to 125K CT with segmentations of up to 104 organs each. The authors also provided interactive visualization tools and made their code publicly available to facilitate further research.

---

### Official Review · Reviewer_ow35 · 2024-07-23
**A different approach to quality control of image segmentation**

**Rating:** 4
**Confidence:** 5
**Clarity:** Yes

**Review:**

This work presents a set of heuristics for the quality control of rib segmentation masks. The idea is novel in that it departs from the current standard in segmentation quality control, which consists in training a machine learning model to predict the quality of the mask.  While the idea is interesting the work fails to demonstrate experimentally that this approach is better.

A second question that remains to be answered is how general these heuristics are. Currently, they are demonstrated in the specific case of rib segmentation. As such, the title and abstract should reflect the scope of the work.

Please see other points below for further detailed comments.

**Strengths:**

Significance
- The paper offers a novel and original solution to the open problem of quality control of medical image segmentation masks.

Relevance
- Quality control of image segmentation masks in the medical domain is a relevant problem to the overall community. This paper proposes a solution that is demonstrated for the specific case of rib segmentation.

Quality
- The proposed heuristics have been demonstrated in a very large set of data, which act as a guarantee of the robustness of the results

**Additional Feedback:**

- How general are these heuristics? In this sense, how can they be applied to other structures seamlessly? For instance, that one related to the volume seems to be very application-specific. What would be needed to adapt and use in another scenario?
- Reference 14 is not a machine learning approach.

**Correctness:**

My understanding is that this paper builds from a previous release of a dataset (references 7 and 8) by defining the heuristics and providing a dashboard with the results of the quality control (QC) analysis. Hence, there is no new dataset or benchmark. The main contribution is the QC of the previously released data.

**Documentation:**

There is enough information regarding the published dashboard

**Limitations:**

Yes, the paper discusses its limitations

**Opportunities For Improvement:**

- The title and abstract should reflect the scope of this paper which lies in the segmentation of ribs from CT images. Currently, these are misleading as they relfect a larger scope (general medical image segmentation), which is not the case. To claim generality, the paper should provide experimental proof that it works in other segmentation tasks.
- The proposed method fails to demonstrate its superiority with respect to machine learning methods for quality control. These latter are currently the state-of-the-art

**Relation To Prior Work:**

- As the main contribution of this work is the definition of heuristics for segmentation mask quality control, the paper fails to provide an experimental comparison with previous works.
- The authors may refer to a recent review [1] that collects multiple previous efforts on quality control of image segmentations, many of which have code ready to be used and which can be used for comparison.


[1] https://doi.org/10.3390/app12083936

**Summary And Contributions:**

This paper addresses the problem of assessing the quality of segmentation masks that may be used for training AI models. Here, a segmentation mask may be the result of user annotation or the output of an AI model that assisted the annotation. Different from most current works that resort to a machine learning model that assesses the quality of the mask, this work defines four heuristics to be used in the context of rib segmentation. The heuristics aim to detect 1) if a region is fully segmented, 2) the presence of multiple components, the 3) laterality of the region and 4) the minimum volume from voxel summation.

This constitutes a novel approach to quality control.

---

> ### Comment · Reviewer_ow35 · 2024-08-19
>
> The ideas proposed in this work offer an alternative approach to quality control of image segmentation. The state-of-the-art at the moment are machine learning based methods. This work does not provide an experimental comparison to them nor a detailed discussion that positions the work wrt to these. The above reasons motivates the rating.

---

### Official Review · Reviewer_VQu4 · 2024-07-25
**Good paper with some problems**

**Rating:** 6
**Confidence:** 5
**Clarity:** Yes

**Review:**

Pros
- The authors have made their code publicly available, which is a strong commitment to reproducibility and transparency. This openness facilitates further research and validation by the community.
- The authors conducted several analyses including left vs. right rib volume consistency, within-patient volume consistency, and comparison to literature. These studies validate the effectiveness of the heuristics from multiple angles.

Cons
- Lack of demographic variation analysis: Even limitation parts have mentioned that the study does not account for potential variations in segmentation accuracy across different demographic groups (e.g., gender, race). But it is recommended to investigate how these heuristic methods perform across different ethnicities, which could provide valuable insights for future research.
- Figure 2 clarity issues: The figure is not clear, and the caption only explains the meaning of the plot on the lower left side. The meaning of the plot on the upper right side is unclear. For example, the first bar is interpreted as using only the connected_volumes filter, but it is not clear why the value equals 7946. The meaning of each bar needs to be explained.
- Population study limitations: The paper does not conduct a population study on organs other than the vertebral segments. It raises the question of whether the volume of vertebral segments is more stable compared to other internal organs such as the pancreas or pancreas, which may vary significantly between individuals. This brings up the concern that the heuristic methods might only be applicable to vertebral segments. The authors should justify this limitation.
- Lack case study from human expert examination, only statistics are shown.

**Strengths:**

See above

**Additional Feedback:**

Overall, I appreciate this work, but there are some minor issues. I will adjust my score based on the authors' rebuttal.

**Correctness:**

Somehow not. The authors do not make the label public and visualize the label, only statistics.

**Documentation:**

Yes

**Ethics:**

This dataset is constructed based on NLST. If author got the permission from NLST, it may not. But authors didn't show that.

**Limitations:**

See above

**Opportunities For Improvement:**

- Conduct demographic variation analysis
- Additional explanation in caption of figure 2
- Justification for population study. For example, conduct population study on other abdomen organ such as kidney and pancreas. Also, it is good to see the effectiveness of this heuristic method in small organs such as adrenal gland.
- Encourage to random select tens of ct scan to be reviewed by human experts.

**Relation To Prior Work:**

Yes

**Summary And Contributions:**

- This paper addresses a critical gap in medical imaging, where manually annotating large-scale medical imaging datasets is prone to variability due to differing clinical conventions.
- Heuristic methods covering various aspects of segmentation quality, including segmentation completeness, connected components, laterality, and minimum volume, are proposed to provide a multifaceted approach to quality assessment.

---

### Author Rebuttal · Authors · 2024-08-16

We thank the reviewers for their very detailed and useful feedback.

Ethics:
- Negative impact of using heuristics: We may filter out patients that should be included in further analysis, which removes viable candidates. We may include patients that should be removed, which could affect comparison to a population, or other segmentation-derived measures. False outliers may not be problematic as our dataset is large.
- Bias and fairness: Though we do have access to gender/race, and could do a qualitative subgroup analysis, we do not have the groundtruth; it will be difficult to assess the bias or segmentation quality of the algorithms. These heuristics could be biased against abnormal body parts/organs, but these may be ones that we want to filter out.
- Data Ethics and Privacy: The NLST data that we used is publicly available through Imaging Data Commons under a CC-BY 4.0 (Creative Commons Attribution 4.0 International License); it was ingested from The Cancer Imaging Archive. The CC-BY 4.0 license allows commercial use without any restrictions. Therefore there were no data privacy or consent issues. Page for confirmation of public access and licensing: https://www.cancerimagingarchive.net/collection/nlst/.
- Clinical and Societal Impact: By using our tools, one can identify outliers and incorrect segmentations. The current version focuses on the ribs, vertebrae etc, but more validation is needed for other organs.
- Stakeholder Engagement: In the future, we will seek opinions of  a radiologist (author on paper) for improving the heuristics, and compare our methods to other outlier-detection based approaches to prove that our method is robust.
- Responsible Development and Deployment: Heuristics will help flag problematic segmentations that warrant a review from radiologists. More heuristics and validation of such heuristics is needed before AI-generated segmentations can be integrated into clinical workflows, without a review from radiologists.
- Broader Ethical Considerations: Users need to know how data was obtained, and variability that arises from these methods, along with performance gaps when the method is applied to other datasets. We follow the FAIR guidelines, and encourage users to use these tools as a starting point for curation and outlier detection on their own data.

Reviewer comments:
- Fitting into track: For the datasets, we provide all of the data and associated metadata that is necessary to perform the quality control [https://github.com/ImagingDataCommons/CloudSegmentatorResults/releases/tag/0.0.1]. We provide a benchmark that people can use to quickly filter out data and detect outliers, maybe it does not fit into the traditional sense of the word “benchmark”. We apologize for the oversight.
- Lack case study from human expert examination: It is difficult to have a qualitative/quantitative review with a large dataset. We can select a small sample and have an expert radiologist provide feedback of specific organs with a manual annotation.
- The authors do not make the label public and visualize the label: We provide the segmentations of the organs of all patients publicly. Our data is hosted in Imaging Data Commons, and is accessible here: https://zenodo.org/records/12004521. Our metadata is released here: https://github.com/ImagingDataCommons/CloudSegmentatorResults/releases/tag/0.0.1.
- Comparison to other ML-based methods: Our approach is rule based and not ML based, but we could consider in future manuscripts comparing against ML based methods as suggested in [1]. However, since those all require training a model with some sort of ground truth/prior, we cannot do this comparison currently.
- Generalizability of heuristics and population study limitations: We first focused on the vertebrae, and ribs, as we wanted to test our methods to what we found in the literature. However, the heuristics we developed could be used for any other structures. The segmentation completeness heuristic and the connected component are not organ specific. The laterality heuristic can only be used for left/right segments, and the minimum volume heuristic also may be applicable only for certain segments. For the veins, one could still use our heuristics (connected components and segmentation completeness). We acknowledge that these may not be sufficient.
- Scope of this paper: We demonstrate not only a rib comparison, but also left vs right kidney volume comparisons (see section 3.3) and also vertebrae analysis (section 3.4). We picked a subset of combinations that we could study and compare to literature – this turned out to be volume. We also made available a dashboard where a user can explore all of these combinations easily, of organs, metrics and heuristics. In the future we can perform more in-depth analysis of more areas.
- Figure 2: Upset plots can be thought of similar to Venn-diagrams [https://ieeexplore.ieee.org/document/6876017] The bars on the left indicate totals, the bars on the top indicate the intersection sizes. The bars on the left show n(A), n(B), and n(C). The bars on the top show all combinations of intersections of A, B, and C. While there are a total of 10147 (n(A ∪ B ∪  C ∪  D)) image series failed on connected components heuristic, only 7946 (n(A)) failed solely by connected components. The remaining 2201 series failed by connected components AND by segmentation completeness check (n(A ∩ B)) or segmentation completeness and volume from voxel summation heuristics (n(A ∩ B ∩ C)).
- References: We believe that reference [14] - “Reverse Classification Accuracy: Predicting Segmentation Performance in the Absence of Ground Truth” is actually a machine learning approach. In 1. Introduction, B. Contribution: “To this end, a classifier is trained using a single image with its predicted segmentation acting as pseudo GT. The resulting reverse classifier (or RCA classifier) is then evaluated on images from a reference database for which GT is available.”

---

### Decision · Program_Chairs · 2024-09-26

**Decision:**

Reject

**Comment:**

This work presents an original and logic-driven set of heuristics to help address the open problem of quality control of medical image segmentation masks. The approach is demonstrated for the specific case of rib segmentation on a large set of data.
The addition of this with the new annotations is important, but significant issues were raised, which don't seem to be fully addressed.
In particular, while the work is interesting and important, it doesn't seem to align with the track that closely. The heuristics are hard to implement, and the code describing them is poorly documented and hard to follow.
The majority of the reviewers request a weak reject, and I concur with that.
If space permits, and the misalignment can be overlooked, I would not be against acceptance as a poster, given that the dataset is of interest, especially if the authors can implement the updates requested.